# Effect of Dietary Supplementation with Omega-3 Fatty Acid on the Generation of Regulatory T Lymphocytes and on Antioxidant Parameters and Markers of Oxidative Stress in the Liver Tissue of IL−10 Knockout Mice

**DOI:** 10.3390/nu16050634

**Published:** 2024-02-24

**Authors:** Daniela Dalpubel Campanari, Ualter Guilherme Cipriano, Thais Fernanda de Campos Fraga-Silva, Leandra Náira Zambelli Ramalho, Paula Payão Ovidio, Alceu Afonso Jordão Júnior, Vânia Luiza Deperon Bonato, Eduardo Ferriolli

**Affiliations:** 1Postgraduate Program in Clinical Medicine, Ribeirao Preto Medical School, University of Sao Paulo, Ribeirao Preto 14049-900, Sao Paulo, Brazil; 2Basic and Applied Immunology Program, Ribeirao Preto Medical School, University of Sao Paulo, Ribeirao Preto 14049-900, Sao Paulo, Brazil; ualtercipriano@usp.br (U.G.C.); vlbonato@fmrp.usp.br (V.L.D.B.); 3Department of Biochemistry and Immunology, Ribeirao Preto Medical School, University of Sao Paulo, Ribeirao Preto 14049-900, Sao Paulo, Brazil; thaisfragasilva@gmail.com; 4Institute of Biological Sciences and Health, Federal University of Alagoas, Maceio 57072-900, Alagoas, Brazil; 5Department of Pathology and Legal Medicine, Ribeirao Preto Medical School, University of Sao Paulo, Ribeirao Preto 14049-900, Sao Paulo, Brazil; lramalho@fmrp.usp.br; 6Department of Health Sciences, Ribeirao Preto Medical School, University of Sao Paulo, Ribeirao Preto 14049-900, Sao Paulo, Brazil; ppayao@usp.br (P.P.O.); alceu@fmrp.usp.br (A.A.J.J.); 7Department of Internal Medicine, Ribeirao Preto Medical School, University of Sao Paulo, Ribeirao Preto 14049-900, Sao Paulo, Brazil; eferriol@fmrp.usp.br

**Keywords:** IL−10^−/−^ mice, inflammation, oxidative stress, *n*-3 PUFA

## Abstract

Introduction: chronic low-grade inflammation, or inflammaging, emerges as a crucial element in the aging process and is associated with cardiovascular and neurological diseases, sarcopenia, and malnutrition. Evidence suggests that omega-3 fatty acids present a potential therapeutic agent in the prevention and treatment of inflammatory diseases, mitigating oxidative stress, and improving muscle mass, attributes that are particularly relevant in the context of aging. The objective of the present study was to evaluate the effectiveness of supplementation with omega-3 fish oil in improving the immune response and oxidative stress in knockout mice for interleukin IL−10 (IL−10^−/−^). Material and methods: female C57BL/6 wild-type (WT) and interleukin IL−10 knockout (IL−10^−/−^) mice were fed during 90 days with a standard diet (control groups), or they were fed/supplemented with 10% of the omega-3 polyunsaturated fatty acid diet (omega-3 groups). Muscle, liver, intestinal, and mesenteric lymph node tissue were collected for analysis. Results: the IL−10^−/−^+O3 group showed greater weight gain compared to the WT+O3 (*p* = 0.001) group. The IL−10^−/−^+O3 group exhibited a higher frequency of regulatory T cells than the IL−10^−/−^ group (*p* = 0.001). It was found that animals in the IL−10^−/−^+O3 group had lower levels of steatosis when compared to the IL−10^−/−^ group (*p* = 0.017). There was even greater vitamin E activity in the WT group compared to the IL−10^−/−^+O3 group (*p* = 0.001) and WT+O3 compared to IL−10^−/−^+O3 (*p* = 0.002), and when analyzing the marker of oxidative stress, MDA, an increase in lipid peroxidation was found in the IL−10^−/−^+O3 group when compared to the IL−10^−/−^ group (*p* = 0.03). Muscle tissue histology showed decreased muscle fibers in the IL−10^−/−^+O3, IL−10^−/−^, and WT+O3 groups. Conclusion: the findings show a decrease in inflammation, an increase in oxidative stress markers, and a decrease in antioxidant markers in the IL−10^−/−^+O3 group, suggesting that supplementation with omega-3 fish oil might be a potential intervention for inflammaging that characterizes the aging process and age-related diseases.

## 1. Introduction

The aging process is intrinsically associated with significant changes in both innate and adaptive immunity, an imbalance between pro- and anti-inflammatory responses, favoring the emergence of low-grade chronic inflammation [1,2]. The inflammaging concept is characterized by the emergence of a chronic pro-inflammatory state in older persons and the progressive increase of serum inflammatory mediators [3,4,5]. Inflammaging is influenced by a diversity of factors, encompassing life experiences, environmental conditions, the intestinal microbiome, physical inactivity, obesity, dietary habits, stress, and exposure to pollutants [6].

The imbalance in the production of free radicals and antioxidants also induces low-grade inflammation and leads to cellular and tissue damage, resulting in greater susceptibility to age-related diseases. Chronic low-grade inflammation and oxidative stress play a crucial role in the development of cardiovascular and neurological diseases, sarcopenia, and malnutrition [1,7,8].

In turn, omega-3, which is a polyunsaturated fatty acid (PUFA), influences gene expression and inflammatory mechanisms, contributing to the regulation of the immune response and the reduction of the production of pro-inflammatory cytokines [9,10]. Studies indicate that omega-3 supplementation benefits cardiovascular health and improves the neuromuscular response to physical training in older people [11,12]. Omega-3 intake is associated with a reduction in the neutrophil-lymphocyte ratio, which suggests a balance between innate and adaptive immune responses, as well as a decreased risk of chronic diseases. Furthermore, omega-3 appears to reduce the adverse effects of the Western diet and to improve inflammation associated with metabolic disorders [13,14]. These studies suggest that omega-3 fatty acids may be promising therapeutic agents for treating age-related diseases such as sarcopenia due to their anti-inflammatory properties and ability to improve oxidative stress [15,16,17].

Interleukin 10 knockout mice (IL−10^−/−^ mice), which exhibit frailty, increased inflammation, reduced muscle strength, and changes in skeletal muscle, may represent an experimental model to study features of age-related diseases [18,19,20]. The understanding of inflammaging in older persons is essential to prevent comorbidities, hospitalizations, and mortality, as well as to develop accurate and effective interventions in the treatment and prevention of age-related diseases [21,22,23]. In the present study, we used IL−10^−/−^ mice to investigate the effectiveness of supplementation with omega-3 fish oil in the reduction of inflammaging and in liver and muscle health. We found that O3 supplementation increases Treg cells in the mesenteric lymph nodes and eliminates lipid accumulation in the liver.

## 2. Materials and Methods

### 2.1. Animals and Diet

Female C57BL/6 wild-type (WT) and homozygous knockout mice for interleukin IL−10 (IL−10^−/−^) 4 weeks old were obtained from the breeding facility of Ribeirao Preto Medical School (FMRP-USP) and from the Department of Immunology at ICB-USP. Animals were maintained under controlled conditions of temperature (22 ± 2 °C) and humidity and in a light cycle (6 a.m.–6 p.m.)/dark (6 p.m.–6 a.m.). Water and food were provided *ad libitum*. The animals were handled in accordance with the recommendations of the Brazilian College of Animal Experimentation, and all procedures were approved by the Ethic Committee (protocol no. 0247/2019) of FMRP-USP (Ribeirao Preto, Sao Paulo, Brazil)

The animals were randomly distributed into four experimental groups. Control groups, WT (*n* = 15) and IL−10^−/−^ (*n* = 13) mice, received an AIN-93 growth diet (containing 75% carbohydrate, 15% protein, and 10% lipid) for 30 days. After that, control groups, WT and IL−10^−/−^, remained fed with the AIN-93 diet, while omega-3 groups, WT+O3 (*n* = 17) and IL10^−/−^+O3 (*n* = 14), were fed/supplemented with a diet rich in fish oil (containing 65% carbohydrate, 15% protein, 10% lipid, and 10% fish oil). Both diets were administered for 90 days; the food was weighed and changed 3 times a week. Supplementation with omega-3 fish oil was purchased from Naturalis^®^ (registration with the Ministry of Health number 4.1480.0006.001-4), and each 1 g of fish oil contained 540 mg of Eicosapentaenoic Acid (EPA) and 100 mg of Docosahexaenoic Acid (DHA) (Naturalis^®^, Arujá, Sao Paulo, Brazil).

The animals were weighed weekly for 90 days. At the end of the experiment, the animals were anesthetized with ketamine (Vetnil, Jacareí, Sao Paulo, Brazil) and xylazine (Syntec, Santana de Parnaíba, Sao Paulo, Brazil) diluted in saline, 100 mg/kg and 10 mg/kg, respectively, administered by intraperitoneal route. Blood was immediately collected by cardiac puncture and centrifuged at 3500 R.P.M at 4 °C for subsequent serum separation. Serum was stored at −80 °C. Liver and muscle tissues were collected, weighed, and frozen at −80 °C; small intestinal tissue was collected and fixed in 10% buffered formalin; and mesenteric lymph nodes were collected and stored for later analysis.

### 2.2. Culture of Mesenteric Lymph Nodes and Determination of IL-6

Mesenteric lymph nodes were dissociated through a cell strainer with the rubber end of a syringe. The remaining cells were washed with RPMI 1640 (Sigma-Aldrich, St. Louis, MO, USA) medium and then adjusted to 1 × 106 cells/mL in RPMI 1640 medium supplemented with 10% heat-inactivated fetal calf serum (Gibco BRL, Grand Island, NY, USA), 2 mM of l-glutamine (Sigma-Aldrich, St. Louis, MO, USA), and 0.1% antibiotic/antimycotic (Sigma-Aldrich, St. Louis, MO, USA). Mesenteric lymph node cell suspensions were distributed in a 48-well plate, stimulated with concanavalin-A (ConA) (20 μg/mL), and incubated at 37 °C in a humidified incubator containing 5% CO_2_. After 48 h of incubation, supernatants were stored at −20 °C for further cytokine quantification. IL-6 was measured in the supernatants of cultures and in the serum samples by enzyme-linked immunosorbent assay using DuoSet (R&D Systems, Minneapolis, MN, USA), following the manufacturer’s instructions. The limit of detection was 15.6–1000 pg/mL.

### 2.3. Determination of Liver Protein

The determination of liver protein was carried out using the commercial kit Biuret method (Labtest Diagnóstica SA, Lagoa Santa, Minas Gerais, Brazil) and readings performed on an Epoch—BioTek (version 2.06) spectrophotometer.

### 2.4. Determination of Hepatic Reduced Glutathione (GSH)

GSH measurement was performed in liver tissue using the method described by Sedlak and Lindsay [24], with minor adaptations. A 50 μL aliquot of the homogenate was mixed with 200 μL of deionized water, 50 μL of 50% trichloroacetic acid (Êxodo Científica, Sumaré, Sao Paulo, Brazil), and 500 μL of 0.02 M EDTA. This solution was stirred in an AP 59—Phoenix Luferco vortex (Araraquara, Sao Paulo, Brazil). After 15 min it was incubated at room temperature and shaken again, then the sample was centrifuged for five minutes at 4000 R.P.M. at 4 °C. After centrifugation, 250 μL of the supernatant were transferred to a new microtube, and 250 μL of TRIS buffer (0.4 M, pH 8.9) were added, along with 25 μL of 0.01 M dithionitrobenzoic acid in methanol (Sigma—D8130). The solution was homogenized, and after five minutes, the absorbance reading was performed at a wavelength of 412 nm on the Epoch—BioTek (version 2.06) spectrophotometer. The concentration was calculated using a standard curve of GSH in EDTA (0.02 M). Blank control was composed of EDTA, TRIS, and DTNB [24].

### 2.5. Determination of Hepatic Vitamin E

Liver analyses of vitamin E (alpha-tocopherol) were performed using a high-performance liquid chromatograph (HPLC), Shimadzu model LC-20AT version 1.25 SP4 (Lynx, Laguna Niguel, CA, USA) a type C-18 column (150 × 4.6 mm–5 µm); and a UV-visible detector model SPD-20A, according to a method adapted from Arnaud and collaborators [25]. In 2 mL microtubes, approximately 100 mg of tissue were homogenized in 0.5 mL of ethanol. Then 0.5 mL of hexane was added and vortexed in AP 59—Phoenix Luferco (Araraquara, Sao Paulo, Brazil) for one minute and centrifuged at 4000 R.P.M. for five minutes in the Sorvall Legend Mach 1.6 R centrifuge. After centrifugation, 100 μL of the hexane phase (supernatant) were transferred to another microtube, where drying was carried out with nitrogen flow, leaving the lipid part of the material in the microtube. 100 μL of the mobile phase (70% acetonitrile, 20% dichloromethane, and 10% methanol) were added and shaken to resuspend the fat. The material was transferred to the vial and analyzed on a Shimadzu HPLC, model LC-20AT, version 1.25 SP4 (Lynx, Laguna Niguel, CA, USA) type C-18 column (250 × 4.6 mm–5 μm), UV-visible detector model SPD-20A. Vitamin E concentration was determined from an alpha tocopherol standard curve [25].

### 2.6. Determination of Hepatic Glutathione Peroxidase (GPx)

To determine GPx, an adaptation of the method described by Paglia and Valentine [26] was carried out. In a 4 mL cuvette, 1.8 mL of phosphate buffer (PBS) and 6.7 μL of sodium azide were added, and then the cuvette was placed in a SpectraMax M3—Molecular Devices spectrophotometer at 37 °C for 30 s. 67 μL of reduced glutathione, 67 μL of NADPH, 2 μL of glutathione reductase, and 67 μL of sample were added to this material at 37 °C. The reaction was started with the addition of 67 μL of hydrogen peroxide (H_2_O_2_). The kinetic reading was taken at 340 nm each 30 s lasting a total of two minutes. GPx concentration was calculated from its molar extinction coefficient [26].

### 2.7. Determination of Hepatic Malondialdehyde (MDA)

The analysis of hepatic MDA was performed according to Erdelmeier and collaborators [27], with minor adaptations. 100 μL of tissue homogenate and 300 μL of 1-methyl-phenylindole solution at a concentration of 10 mM in acetonitrile and methanol (2:1, *v*/*v*) were placed in a microtube, which was then shaken in an AP 59—Phoenix Luferco vortex (Araraquara, Sao Paulo, Brazil). Subsequently, 75 μL of fuming HCl (37%) was added and vortexed again. The samples were then incubated in a water bath at 45 °C for 40 min and centrifuged at 4000 R.P.M. at 4 °C for five minutes. The absorbance reading of the supernatant was performed using Epoch—BioTek (version 2.06) equipment at a wavelength of 586 nm, and the MDA concentration was calculated by comparing it to an MDA standard curve [27].

### 2.8. Determination of the Weight of Muscle and Liver Tissues

Both the liver and the tibialis anterior, soleus, gastrocnemius, and extensor digitorum longus muscles were excised and weighed on a precision scale to determine their mass.

### 2.9. Histopathological Analysis

Histopathological analysis was performed as described by Prado and collaborators [28]. Samples from the liver, anterior tibialis muscle, and small intestine were collected and fixed in 4% paraformaldehyde in PBS for 24 h, with subsequent processing for inclusion in paraffin and stained with hematoxylin and eosin (H&E). For liver tissue, samples from the right lobe were collected to determine the degree of steatosis from zero to 3, adapted from Kleiner and collaborators [29], with grade 0 being up to 5%; grade 1: 5 to 33%; grade 2: 33 to 66%; and grade 3: above 66% [29]. For the tibialis anterior muscle, the analysis was carried out using the thickness of muscle fibers, where there was a separation of normal and slightly decreased. For the analysis of the small intestine, there was a separation of those with and without changes in intestinal crypts and microvilli [30].

### 2.10. Immunophenotyping of Mesenteric Lymph Node Cells

Dissociated mesenteric lymph node cells were incubated with the following fluorochrome-labeled antibodies (BD Horizon): anti-CD3 FITC (clone 145-2C11) and anti-CD4 APC (clone RM4–5) for the evaluation of T lymphocytes by flow cytometry. For the evaluation of regulatory T cells (Treg), cells were analyzed by extracellular expression of CD25 (anti-CD25 PerCP Cy5.5, clone PC61) and intranuclear expression of FOXP3 (anti-Foxp3 PE, clone FJK-16s) in T cells (CD3+CD4+ cells). After labeling, the cells were washed and fixed in 1% paraformaldehyde. Data acquisition was carried out using the FACS Melody flow cytometer (BD Biosciences, San Jose, CA, USA) at the Department of Biochemistry and Immunology (FMRP-USP, Ribeirao Preto, SP, Brazil), and the data was analyzed with the FlowJo software version 10 (Becton Dickinson and Company, Franklin Lakes, NJ, USA). Cells were gated by morphology for lymphocytes (FSC-A vs. SSC-A), singlets, double positive for CD3 and CD4, followed by CD25^+^ and then FOXP3^+^ cells, as represented in Appendix A.

### 2.11. Statistical Analysis

The variables studied were compared between the groups by Kruskal-Wallis, because they are non-parametric data, with a Bonferroni post-hoc test when applicable. Only the variable FOXP3 was compared by ANOVA with a Bonferroni post-hoc test since the variable was submitted to a log_10_ transformation before the comparison by ANOVA (compositional data). All analysis and graphs were performed by IBM SPSS version 25.0 (IBM Corp., Armonk, NY, USA), and bilateral values of *p* < 0.05 were considered significant differences.

## 3. Results

### 3.1. Effects of PUFAs on Body Weight Gain and Energy Intake

When analyzing the body weight of the animals after 90 days of supplementation, it was observed that there was a greater weight gain at the end of the experiment in the omega-3 groups when compared to the group that was not supplemented (Table 1a). Mice supplemented with O3, WT, or IL−10^−/−^, exhibited a significant increase in weight compared to their respective control groups (WT or IL−10^−/−^). The IL−10^−/−^+O3 group had a significantly increased weight compared to the WT+O3 group (*p* = 0.013) (Table 1b). We also observed that IL−10^−/−^ animals without supplementation gained significantly more weight compared to WT animals without supplementation.

When analyzing the weight variation delta, the difference between the initial and final value of the animals’, we also observed that the groups that received supplementation had greater weight compared to the groups that did not receive supplementation: WT and WT+O3 (*p* = 0.006), IL−10^−/−^ and IL−10^−/−^+O3 (*p* = 0.027) (Table 1b). When analyzing the final weight, we found a difference between WT and WT+O3 (*p* = 0.036).

When evaluating the amount of food consumed, intake was similar between the groups and showed no statistical difference.

### 3.2. Assessment of Intestinal Tissue and Immunophenotyping of Cells from the Mesenteric Lymph Node

The morphometric histological parameters of the small intestine were evaluated, and no difference was found among groups when comparing intestinal villi and crypts (Figure 1).

When we cultured mesenteric lymph node cells with concanavalin (ConA) to access the inflammatory profile in the secondary lymphoid organs of the intestine, we found a non-significant decrease in the production of IL-6 in the culture supernatants of supplemented groups, WT or IL−10^−/−^, animals compared to their respective non-supplemented controls (Appendix A).

To understand whether the supplementation with omega-3 increased the frequency of regulatory T cells (Treg), important for suppressing the inflammatory response, we performed the phenotype of lymphocytes in the mesenteric lymph nodes and evaluated the frequency of Treg cells (Figure 2). The gate strategy used to evaluate CD3^+^CD4^+^CD25^+^FOXP3^+^ cells is depicted in Appendix A.

We found a higher frequency of FOXP3^+^ CD4^+^ lymphocytes in the IL−10^−/−^+O3 group compared to the IL−10^−/−^ group (*p* = 0.001). Although not significant, the WT+O3 group also exhibited an increase in the frequency of Treg cells compared to the WT group (*p* = 0.350). The difference was also observed in the WT groups compared to IL−10^−/−^ (*p* = 0.026) and WT+O3 with IL−10^−/−^ (*p* = 0.001) (Figure 3).

We also see that there was an increase in Foxp3 expression in supplemented IL−10^−/−^ animals when compared to non-supplemented IL−10^−/−^ animals, which obtained similar results to WT animals, suggesting an increase in Treg cells.

### 3.3. Assessment of Liver Tissue, Antioxidant Function, and Markers of Oxidative Stress

Histological analysis of liver tissue (Figure 4) showed lower levels of steatosis in the animals of the IL−10^−/−^+O3 group than the IL−10^−/−^ group (*p* = 0.017), suggesting a possible protection for steatosis in the IL−10^−/−^ mice supplemented with omega-3 (Table 2b). No significant differences were found comparing the other groups. We observed that even though there was no statistical difference between the WT+O3 and WT groups, the WT+O3 animals presented lower levels of steatosis score (Table 2a). However, when examining liver tissue weight, there were no significant differences between groups.

No changes were observed in the antioxidant function of GSH and GPX (Table 3). The findings showed greater vitamin E, which has an important antioxidant function protecting adipose tissue from free radical activity [31], WT +O3 compared to IL−10^−/−^+O3 (*p* = 0.002). However, no difference was found between IL−10^−/−^ and IL−10^−/−^+O3 (Figure 5a and Table 3b).

Because lipid peroxidation constitutes a chain reaction of polyunsaturated acids in the cell membrane, generating free radicals that alter their permeability, fluidity, and integrity [8], we analyzed the oxidative stress marker MDA. We observed an increase in the lipid peroxidation in the IL−10^−/−^ +O3 group compared to the IL−10^−/−^ group (*p* = 0.03) (Figure 5b and Table 3b).

### 3.4. Tissue Assessment: Anterior Tibialis, Long Digital Extensor, Soleus, and Gastrocnemius Muscles

The analysis of muscle weight showed no significant difference among the groups (Appendix A).

The GPX enzyme acts in the control of free radical formation [32]. However, no changes were observed in the GPX antioxidant function of muscle tissue (Appendix A).

Muscle tissue histology (Figure 6) showed decreased muscle fibers in the IL−10^−/−^+O3, IL−10^−/−^, and WT+O3 groups. Both the IL−10^−/−^+O3 and IL−10^−/−^ groups did not show significant differences, while the WT+O3 and WT groups were significantly different (*p* = 0.001) (Table 4).

## 4. Discussion

This study analyzed the Treg cells, IL-6 secretion, liver, and muscle responses in IL−10^−/−^ mice after a 90-day supplementation with fish oil rich in omega-3. The results show that the supplemented animals had a higher frequency of Foxp3^+^ CD4^+^ (Treg) cells in mesenteric lymph nodes, less hepatic steatosis, higher levels of MDA, lower levels of vitamin E, and differences in the size of muscle fibers compared to non-supplemented IL−10^−/−^ mice.

The supplemented animals had the highest body weight seen on the last day of the experiment. When comparing the delta value in the animals’ weight gain, the groups that received supplementation also showed greater weight gain throughout the experiment compared to the non-supplemented groups. It is worth mentioning that the duration and amount of lipids added to the diet of supplemented animals may explain differences between the animals’ weight gain in relation to previous studies, since when there is an increase in the amount of lipids, animals tend to gain weight [33,34].

IL−10^−/−^ group gained more weight compared to WT animals; this result may have been due to the fact that the animal is knockout and has low-grade chronic inflammation, making it more susceptible to fat accumulation [35], which shows that low-grade chronic inflammation is a critical factor implicit in obesity as it can cause significant changes in the intestinal barrier, resulting in a two- to three-fold increase in serum LPS concentrations [36]. Similarly, visceral adipose tissue and inflammation are hallmarks of obesity in mouse models, characterized by increased numbers of macrophages and other immune system cells attributed to macrophage residential expansion and increased recruitment [37], resulting in higher levels of pro-inflammatory cytokine production [37,38].

Intestinal histology showed no changes among groups regarding intestinal villi and crypts. It is worth mentioning that a diet rich in PUFA improves intestinal villi and crypts in injured tissue [39,40], which did not occur in the present study since the intestinal tissue remained intact throughout the experiment; therefore, there may be no improvement in tissue already intact.

It is noteworthy that even without finding a significant difference between the groups in the production of IL-6 in the mesenteric lymph node culture, in the supplemented IL−10^−/−^ group there was a lower production of the cytokine. The lower levels of IL-6 were associated with the high expression of FOXp3, also detected in the mesenteric lymph nodes of the same groups of animals that received supplementation, suggesting that increased intake of omega-3 fatty acids in the diet can beneficially modulate the composition of the immune system cell population. Corroborating our findings, Camacho-Muñoz and collaborators found in their study with animal models that a diet rich in omega-3 reduces the concentration of long-chain ceramides (C23–C26) in plasma and adipose tissues and increases the prevalence of CD4+Foxp3+CD25+ Treg cells in lymphoid organs, supporting the notion that ceramide-mediated pathways sustain Treg differentiation [13].

Animals belonging to the IL−10^−/−^+O3 group showed a significant reduction in hepatic steatosis compared to their non-supplemented counterparts. It was observed that mice supplemented with PUFA exhibit greater preservation of liver tissue integrity when evaluated in relation to steatosis. The hepatic protective capacity conferred by *n*-3 PUFA fatty acids against hepatic steatosis is attributed to the protective effects of the FFA4 receptor on hepatocytes as well as n-3 PUFA metabolites such as resolvins and protectins. Furthermore, the anti-inflammatory effect of the FFA4 receptor on macrophages and adipocytes [41,42].

There was no difference in the activity of GSH and GPx among the groups of animals, however, greater vitamin E activity was found in the groups of non-supplemented animals compared to their respective supplemented groups. It is important to highlight that α-tocopherol, a form of vitamin E, plays a crucial role in protecting the integrity of the cell membrane against oxidative damage in collaboration with other antioxidants, such as vitamin C and GPx. The present study indicates that antioxidant activities were lower in supplemented animals, which can be attributed to the use of these antioxidants to contain the development of hepatic steatosis since omega-3, by itself, exerts a protective effect against free radicals, reducing the need for high levels of antioxidants [43,44].

Additionally, there was an increase in MDA levels, a marker of oxidative stress, in the IL−10^−/−^+O3 group compared to the IL−10^−/−^ group, while the WT groups did not show significant differences in MDA levels. It is worth remembering that MDA is the result of lipid peroxidation, therefore, when there is an increase in PUFA consumption, there is an increase in MDA [45]. Consumption in humans should be cautious and its extrapolation viewed with care, considering the amount administered and metabolic differences between individuals [46]. The high consumption of antioxidants by individuals who exercise regularly and athletes can lead to potential negative effects [47]. Studies indicate that prolonged administration of high doses of antioxidants inhibits redox-sensitive signaling pathways, reducing exercise-induced physiological adaptations such as increased insulin sensitivity, mitochondrial biogenesis, and muscle hypertrophy [48,49].

There was no difference between the groups when analyzing the weight of the anterior tibialis, long digital extensor, soleus, and gastrocnemius muscles, although in the supplemented animals there was an increase in the body weight. It is worth mentioning that the growth and hypertrophy of skeletal muscle depend on the proliferation of satellite cells, as resistance exercises induce muscle hypertrophy through their activation and proliferation, subsequent chemotaxis, and fusion of these to pre-existing muscle fibers [50], and in the present study there was no strength and/or resistance training.

Regarding histology, decreased muscle fibers were found in all groups, except in the non-supplemented WT group. It is known that the anabolic response of PUFA occurs through mechanisms involved in protein synthesis, with the positive regulation of the Akt-mTOR-p70S6k pathway [51,52]. Smith and colleagues demonstrated in healthy human models that 8-week EPA/DHA supplementation increased mTOR and p70s6k phosphorylation in muscle biopsies but found no effect on Akt [53]. Mice on a high-fat diet for 20 weeks exhibited stress and reductions in the activity of the mTOR pathway [54]. Although our results did not show a role for omega-3s or FADS2 in protein synthesis, we provide evidence that n-3s may have an effect on skeletal muscle protein degradation when supplemented over a long period of time.

## 5. Conclusions

This study shows that lipid modulation in a diet containing omega-3 polyunsaturated fatty acids was able to induce a series of metabolic effects, including an increase in regulatory T cells and an improvement in hepatic steatosis, but demonstrated an increase in oxidative stress levels and a decrease in antioxidant functions. Therefore, the use of omega-3 polyunsaturated fatty acids can be a low-cost therapeutic alternative, as long as they are dosed with caution and under the supervision of a professional capable of understanding their benefits.

## Figures and Tables

**Figure 1 nutrients-16-00634-f001:**
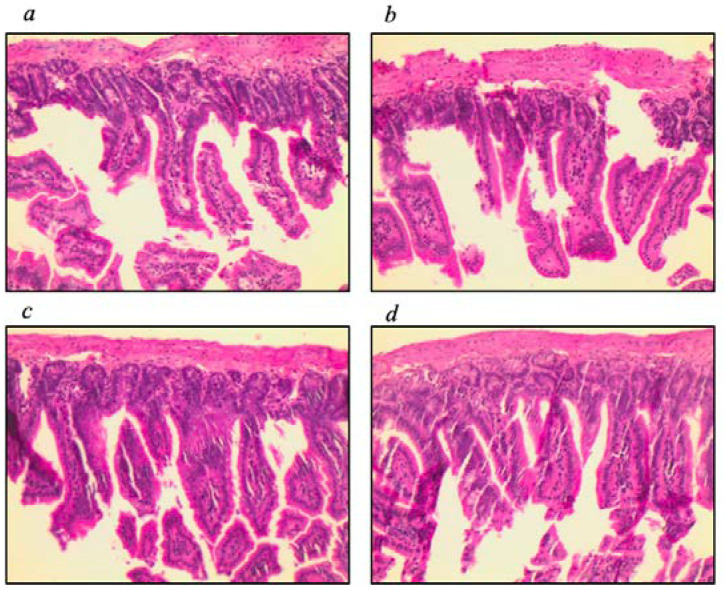
Histology of the small intestine. Histology of the medial small intestine of the groups (**a**) IL−10^−/−^+O3; (**b**) IL−10^−/−^; (**c**) WT+O3; and (**d**) WT. Magnification: ×40.

**Figure 2 nutrients-16-00634-f002:**
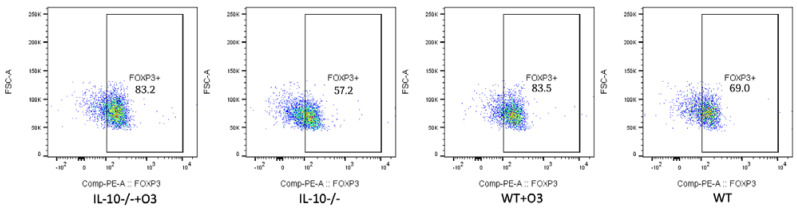
Representative flow cytometry analysis of FoxP3 expression. Cells derived from mesenteric lymph nodes of the groups IL−10^−/−^+O3; IL−10^−/−^; WT+O3 and WT were cultured with ConA and evaluated after 48 h by flow cytometry.

**Figure 3 nutrients-16-00634-f003:**
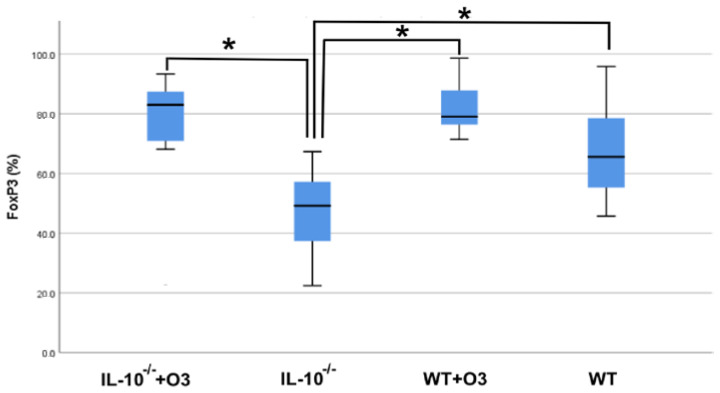
FOXP3 expression in T cells. Cells derived from mesenteric lymph nodes of the groups IL−10^−/−^+O3; IL−10^−/−^; WT+O3 and WT were cultured with ConA and evaluated after 48 h by flow cytometry. Results are representative of three independent experiments: IL−10^−/−^+O3 (*n* = 14); IL−10^−/−^ (*n* = 13); WT+O3 (*n* = 17), and WT (*n* = 15). The bars show the significant differences between groups (* = *p* < 0.05).

**Figure 4 nutrients-16-00634-f004:**
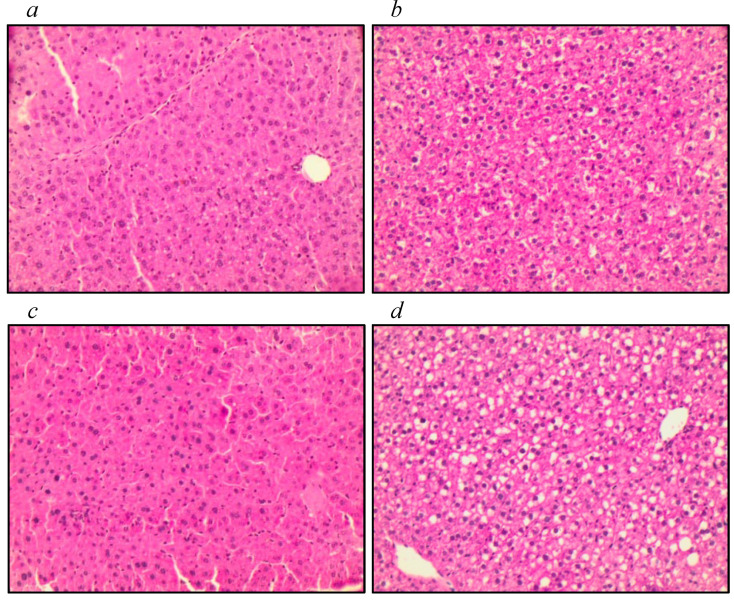
Histology of liver tissue. Histology of the right liver lobe of groups (**a**) IL−10^−/−^ +O3; (**b**) IL−10^−/−^; (**c**) WT+O3; and (**d**) WT. Magnification: ×40.

**Figure 5 nutrients-16-00634-f005:**
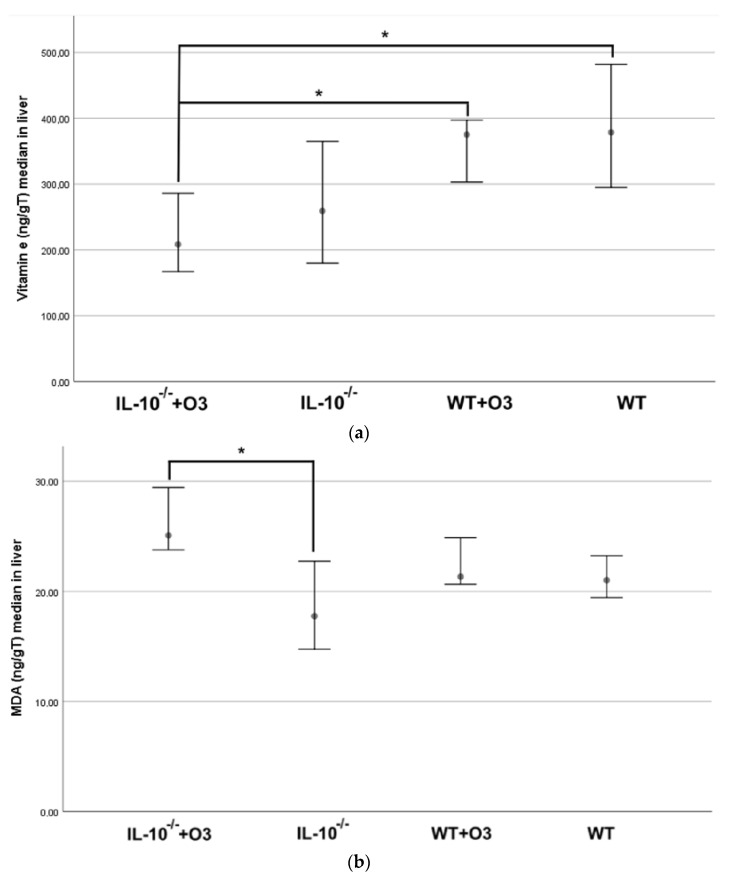
(**a**,**b**) Antioxidant function of Vitamin E and oxidative stress by MDA of IL−10^−/−^+O3; IL−10^−/−^; WT+O3 e WT groups. Results are representative of three independent experiments: IL−10^−/−^ +O3 (*n* = 14); IL−10^−/−^ (*n* = 13); WT+O3 (*n* = 17); and WT (*n* = 15). The bars show the significant differences between groups (* = *p* < 0.05).

**Figure 6 nutrients-16-00634-f006:**
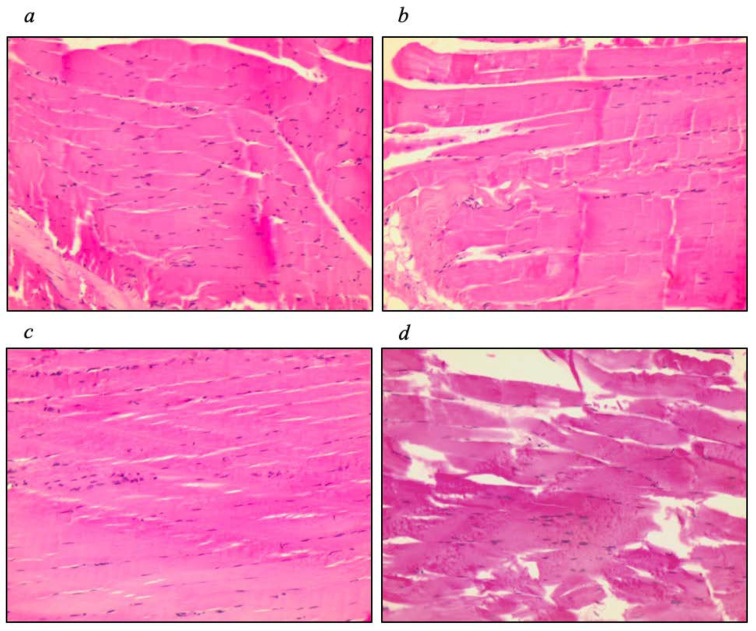
Histology of muscle tissue. Histology of the anterior tibial muscle of groups (**a**) IL−10^−/−^ +O3; (**b**) IL−10^−/−^; (**c**) WT+O3; and (**d**) WT. Magnification: ×40.

**Table 1 nutrients-16-00634-t001:** (**a**,**b**) Animal weight. Distribution of initial, final, and delta weights between groups IL−10^−/−^ +O3; IL−10^−/−^; WT+O3 e WT.

(**a**)								
**Variables**	**IL−10^−/−^+O3**		**IL−10^−/−^**		**WT**		**WT+O3**	
	**Median**	**IQR**	**Median**	**IQR**	**Median**	**IQR**	**Median**	**IQR**
Starting weight	22.15	21.3–22.5	22.0	21.4–22.8	19.3	18.7–19.7	19.5	18.5–20.2
Final weight	29.65	27.4–30.5	26.4	24.9–27.6	25.5	24–26.4	23.2	21.5–23.8
Weight delta	7.45	5.5–8.7	4.3	3.5–4.9	6.5	4.9–8.2	3.9	2.2–4.7
(**b**)								
Bonferroni post-hoc test *p*-Value								
	IL−10^−/−^ vs. Il-10^−/−^+O3	IL−10^−/−^ vs. WT	IL−10^−/−^ vs. WT+03		IL−10^−/−^+O3 vs. WT		IL−10^−/−^+O3 vs. WT+03	WT vs. WT+O3
Starting weight	>0.999	0.001 *	0.001 *		<0.001 *		<0.001 *	>0.999
Final weight	0.251	0.004 *	>0.999		<0.001 *		0.013 *	0.036 *
Weight delta	0.027 *	>0.999	>0.999		<0.001 *		>0.999	0.006 *

Note: IQR means interquartile range. * indicates statistical significance (*p* < 0.05).

**Table 2 nutrients-16-00634-t002:** (**a**,**b**) Steatosis score. Steatosis score with median and interquartile range between groups IL−10^−/−^ +O3; IL−10^−/−^; WT+O3 e WT.

(**a**)									
**Variables**	**IL−10^−/−^+O3**		**IL−** **10^−/−^**		**WT+O3**		**WT**		**Kruskal-Wallis** ** *p* ** **-Value**
	**Median**	**IQR**	**Median**	**IQR**	**Median**	**IQR**	**Median**	**IQR**	
Scores	1.0	0–1.0	3.0	1.50–3.0	1.0	0–10	2.0	0–3.0	0.009 *
(**b**)									
Bonferroni post-hoc test *p*-Value									
	IL−10^−/−^ vs. Il-10^−/−^+O3		IL−10^−/−^ vs. WT		IL−10^−/−^ vs. WT-03		IL−10^−/−^+O3 vs. WT	IL−10^−/−^+O3 vs. WT+03	WT vs. WT+O3
Scores	0.017 *		0.260		0.017 *		>0.999	>0.999	>0.999

Note: IQR means interquartile range. * indicates statistical significance (*p* < 0.05). The Bonferroni post-hoc test is not applicable when the Kruskal-Wallis *p*-value is >0.05.

**Table 3 nutrients-16-00634-t003:** (**a**,**b**) Antioxidant function and oxidative stress of liver tissue. Glutathione peroxidase, reduced glutathione, vitamin E, and malondialdehyde from the groups IL−10^−/−^+O3; IL−10^−/−^; WT+O3 e WT.

(**a**)														
**Variables**	**IL−10^−/−^+O3**			**IL−10^−/−^**			**WT+O3**			**WT**				**Kruskal-Wallis** ** *p* ** **-Value**
	**Median**		**IQR**	**Median**	**IQR**		**Median**	**IQR**		**Median**		**IQR**		
Liver weight	1.2		1.1–1.3	1.2	1.2–1.3		1.1	1.1–1.2		1.2		1.1–1.2		0.208
GSH hepatic	201.7		167.4–269.6	216.2	171.2–271.1		270.0	232.7–278.7		231.0		197.6–320.0		0.236
GPx hepatic	224.7		148.9–239.3	237.5	219.4–263.5		217.8	201.1–240.2		223.0		207.3–236.6		0.257
MDA hepatic	25.0		24.0–29.0	17.7	15.6–24.8		21.3	20.6–24.8		21.0		19.4–23.2		0.025 *
Vit E hepatic	208.2		167.4–277.5	259.0	185.6–336.1		374.9	302.8–397.1		378.5		294.7–481.7		<0.001 *
(**b**)														
Bonferroni Post-hoc test *p*-Value														
		IL−10^−/−^ vs. IL−10^−/−^+O3		IL−10^−/−^ vs. WT		IL−10^−/−^ vs. WT+03			IL−10^−/−^+O3 vs. WT		IL−10^−/−^+O3 vs. WT+03		WT vs. WT+O3	
MDA hepatic		0.030 *		>0.999		>0.999			0.085		0.236		>0.999	
VIT E hepatic		0.951		0.139		0.211			0.001 *		0.002*		>0.999	

Note: IQR means interquartile range. * indicates statistical significance (*p* < 0.05). The Bonferroni post-hoc test is not applicable when the Kruskal-Wallis *p*-value is >0.05.

**Table 4 nutrients-16-00634-t004:** Muscle tissue score. Tibialis anterior muscle score of IL−10^−/−^ +O3; IL−10^−/−^; WT+O3 e WT.

		Groups								Chi-Square *p*-Value	Bonferroni Post-Hoc Test	
		IL−10^−/−^+O3		IL−10^−/−^		WT+O3		WT			IL−10^−/−^ vs. WT	WT vs. WT+O3
		*n*	%	*n*	%	*n*	%	*n*	%			
Muscle	Slender	9	64.3%	11	84.6%	15	88.2%	2	15.4%	<0.001 *	0.002 *	<0.001 *
	Normal	5	35.7%	2	15.4%	2	11.8%	11	84.6%			

Notes: * indicates statistical significance (*p* < 0.05). The other comparisons between groups were not statistically significant (*p* < 0.05).

## Data Availability

Data are contained within the article and Appendix A.

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
