# Peer review of "Effect of Dietary Supplementation with Omega-3 Fatty Acid on the Generation of Regulatory T Lymphocytes and on Antioxidant Parameters and Markers of Oxidative Stress in the Liver Tissue of IL−10 Knockout Mice"

_nutrients, 2024, doi:10.3390/nu16050634_

Round 1

Reviewer 1 Report

Comments and Suggestions for Authors

Dear Authors, 

This is an interesting work assessing the role of omega-3 supplementation on IL-10 knock-out mice. I have questions mostly related to the material and methods section: 

- Animals. It is not included how many animals were used WT and knock-out, neither on the experimental groups. Please, include this, as it is important to understand why there were not significant differences in some of the results (maybe there were not enough animals per experimental group to see any differences?). 

- Why did you choose the concanavalin technique to determine the levels of IL-6? Would it be possible to measure it by ELISA or Western blot? 

- You measured antioxidant enzymes, not oxidative stress specifically. Would it be possible to measure oxidative stress? 

- Statistical analysis. Please, indicate why you used Kruskal-Wallis for the majority of the variables vs. ANOVA for only one variable. I assume it was because the majority of the variables were not parametric, only FOXP3, parametric. Please, explain on this section. 

Thank you very much. 

Author Response

Nutrients (ISSN 2072-6643): Effect of dietary supplementation with omega-3 fatty acid on the generation of regulatory T lymphocyte and on antioxidant parameters and markers of oxidative stress in liver tissue of IL-10 knockout mice.

Authors' comments: 

The authors are grateful for the thorough reading, analysis and questioning made by the reviewers. Most of the questions and suggestions were accepted and deeply appreciated. Your considerations were no doubt pertinent; they allowed us to present a more coherent manuscript. The main changes resulting from their input included a meticulous reconsideration of the original manuscript.

Please, find below the answers to each suggestion made by the reviewers. All modifications are highlighted in yellow in the new version of the manuscript.

Reviewer 1

Dear Authors, 

This is an interesting work assessing the role of omega-3 supplementation on IL-10 knock-out mice. I have questions mostly related to the material and methods section: 

- Animals. It is not included how many animals were used WT and knock-out, neither on the experimental groups. Please, include this, as it is important to understand why there were not significant differences in some of the results (maybe there were not enough animals per experimental group to see any differences?). 

Response: Sorry for this. We included this information in Methods section. Please see lines 99, 100, 101.

- Why did you choose the concanavalin technique to determine the levels of IL-6? Would it be possible to measure it by ELISA or Western blot? 

Response: Concanavalin A is a mannose/glucose-binding lectin isolated from Jack beans (Canavalia ensiformis) that can activate the immune system, recruit lymphocytes and elicit cytokine production [1] and has been reported to activate NFAT (nuclear factor of activated T cells), a family of transcription factors that are important in the development and function of the immune system, including T cell receptor (TCR) engagement [2]. We choose the concanavalin to evaluated T cell activation and not specific to IL-6 production. However, we analyzed different cytokines and presented the results of those that were detectable.

  1. Dwyer JM. & Johnson C., 1981. The use of concanavalin A to study the immunoregulation of human T cells. Clin Exp Immunol. 46(2): 237–249.
  2. Bemer V. & Truffa-Bachi P., 1996. T cell activation by concanavalin A in the presence of cyclosporin A: immunosuppressor withdrawal induces

- You measured antioxidant enzymes, not oxidative stress specifically. Would it be possible to measure oxidative stress? 

Response: Thanks for the comment. Oxidative stress is also assessed through the quantification of antioxidant substances (enzymatic and non-enzymatic), since low levels may indicate the installation of the process and also through lipid peroxidation [1,2].

  1. Mecrury, S.M.; Gorden, D.;Wilson, R.; Bradley, H.; Gemmel, C.G.; Patterson, J.R.; Rumally, A.G.; Maccuish, A.C. A comparison of different methods of assaying free radical activity in type II diabetes and peripheral vascular disease. Diabetic Medicine, v.10, p.331-335, 1993.
  2. Vijayaraghavan, R.; Suribabu, C.S.; Sekar, B.; Oommen, P.K.; Kavithalakshmi, S.N.; Madhusudhanan, N.; Panneerselvam C. Protective role of vitamin E on the oxidative stress in Hansen’s disease (Leprosy) patients. European Journal of Clinical Nutrition, v.59, p.1121–1128, 2005.

- Statistical analysis. Please, indicate why you used Kruskal-Wallis for the majority of the variables vs. ANOVA for only one variable. I assume it was because the majority of the variables were not parametric, only FOXP3, parametric. Please, explain on this section. 

Response: Sorry for this. We included this information in Methods section. Please see lines 171 and 172.

Reviewer 2 Report

Comments and Suggestions for Authors

The title should be more precise. For example: “Dietary supplementation with omega-3 fatty acid induces a decrease in local inflammation, both in mesenteric lymph nodes and in hepatic steatosis, but an increase in oxidative stress levels and a decrease in antioxidant functions in liver tissue of IL- 4 10 knockout mice”.

In the Introduction, the following references should be added:

1) Role of the orexin system on arousal, attention, feeding behaviour and sleep disorders.

Acta Medica Mediterranea 33(4):645-649, 2017.

DOI:10.19193/0393-6384_2017_4_096

2) Effects of an high-fat diet enriched in lard or in fish oil on the hypothalamic amp-activated protein kinase and inflammatory mediators.

Frontiers in Cellular Neuroscience.

Volume 10, Issue JUN 9 June 2016.

DOI: 10.3389/fncel.2016.00150.

In the Discussion, caution in the use of omega 3 polyunsaturated fatty acids should be widely reported and well underlined.

Author Response

Nutrients (ISSN 2072-6643): Effect of dietary supplementation with omega-3 fatty acid on the generation of regulatory T lymphocyte and on antioxidant parameters and markers of oxidative stress in liver tissue of IL-10 knockout mice.

Authors' comments: 

The authors are grateful for the thorough reading, analysis and questioning made by the reviewers. Most of the questions and suggestions were accepted and deeply appreciated. Your considerations were no doubt pertinent; they allowed us to present a more coherent manuscript. The main changes resulting from their input included a meticulous reconsideration of the original manuscript.

Please, find below the answers to each suggestion made by the reviewers. All modifications are highlighted in yellow in the new version of the manuscript.

Reviewer 2

The title should be more precise. For example: “Dietary supplementation with omega-3 fatty acid induces a decrease in local inflammation, both in mesenteric lymph nodes and in hepatic steatosis, but an increase in oxidative stress levels and a decrease in antioxidant functions in liver tissue of IL- 4 10 knockout mice”.

Response: Sorry for this. During submission, the system placed the first sentence of the introduction in the title of the article. We contacted the journal and changed it, the title is: “Effect of dietary supplementation with omega-3 fatty acid on the generation of regulatory T lymphocyte and on antioxidant parameters and markers of oxidative stress in liver tissue of IL-10 knockout mice”.

In the Introduction, the following references should be added:

1) Role of the orexin system on arousal, attention, feeding behaviour and sleep disorders. Acta Medica Mediterranea 33(4):645-649, 2017.

DOI:10.19193/0393-6384_2017_4_096

2) Effects of an high-fat diet enriched in lard or in fish oil on the hypothalamic amp-activated protein kinase and inflammatory mediators. Frontiers in Cellular Neuroscience. Volume 10, Issue JUN 9 June 2016. DOI: 10.3389/fncel.2016.00150.

Response: Thank you for the suggestion. We included the suggested references 2. Please, see lines 69. However, after careful consideration, we realized that the inclusion of suggested article 1 may compromise the cohesion and logic of my work, even though it is an extremely rich article with a very interesting topic.

In the Discussion, caution in the use of omega 3 polyunsaturated fatty acids should be widely reported and well underlined.

Response: Thank you for the suggestion, we emphasize this information in the discussion. Please, see lines 346-352.

Reviewer 3 Report

Comments and Suggestions for Authors

This study is original in investigating the effect of dietary supplementation with omega-3 fatty acid on the generation of regulatory T lymphocyte and on antioxidant parameters and markers of oxidative stress in liver tissue of IL-10 knockout mice. It addresses an important role in the prevention and treatment of inflammatory diseases, and when the objectives are achieved, the results will contribute to and advance scientific knowledge and the application of omega-3 fatty acids for reducing cancer risk. However, there are deficiencies in reporting the study, for example, the experiments, methods, and analysis to address the objectives were not clearly described. In addition, there is not enough background information for how the analytical results were interpreted that led to the conclusions, which, in the current form, are lacking substantial support and confidence. This manuscript needs additional information before being considered for publication

1) The author should check the whole manuscript for any grammatical errors and any other differences.

2) Line 191, change “p” to “p. Please check this manuscript to avoid similar errors.

3) Line 131, For each instrument or software, you have to provide information regarding the model, company etc....

4) Figures 2: increase the font size and the quality of the graphs. Please check this manuscript to avoid similar errors.

Comments on the Quality of English Language

 Moderate editing of English language required

Author Response

Nutrients (ISSN 2072-6643): Effect of dietary supplementation with omega-3 fatty acid on the generation of regulatory T lymphocyte and on antioxidant parameters and markers of oxidative stress in liver tissue of IL-10 knockout mice.

Authors' comments: 

The authors are grateful for the thorough reading, analysis and questioning made by the reviewers. Most of the questions and suggestions were accepted and deeply appreciated. Your considerations were no doubt pertinent; they allowed us to present a more coherent manuscript. The main changes resulting from their input included a meticulous reconsideration of the original manuscript.

Please, find below the answers to each suggestion made by the reviewers. All modifications are highlighted in yellow in the new version of the manuscript.

Reviewer 3

This study is original in investigating the effect of dietary supplementation with omega-3 fatty acid on the generation of regulatory T lymphocyte and on antioxidant parameters and markers of oxidative stress in liver tissue of IL-10 knockout mice. It addresses an important role in the prevention and treatment of inflammatory diseases, and when the objectives are achieved, the results will contribute to and advance scientific knowledge and the application of omega-3 fatty acids for reducing cancer risk. However, there are deficiencies in reporting the study, for example, the experiments, methods, and analysis to address the objectives were not clearly described. In addition, there is not enough background information for how the analytical results were interpreted that led to the conclusions, which, in the current form, are lacking substantial support and confidence. This manuscript needs additional information before being considered for publication.

1) The author should check the whole manuscript for any grammatical errors and any other differences.

Response: Sorry for this. The whole manuscript was check for any grammatical errors.

2) Line 191, change “p” to “p”. Please check this manuscript to avoid similar errors.

Response: Sorry for this. We changed the “p” for italic form.

3) Line 131, For each instrument or software, you have to provide information regarding the model, company etc....

Response: As suggested, we provided information regarding model and company. Please see lines 132.

4) Figures 2: increase the font size and the quality of the graphs. Please check this manuscript to avoid similar errors.

Response: As suggested, we increased the font size and the quality of the graph.
